# Defining predictors of responsiveness to advanced therapies in Crohn's disease and ulcerative colitis: protocol for the IBD-RESPONSE and nested CD-metaRESPONSE prospective, multicentre, observational cohort study in precision medicine

Nicola J Wyatt [1,2] Hannah Watson [1] Carl A Anderson [3] Nicholas A Kennedy [4,5] Tim Raine [6] Tariq Ahmad [4,5] Dean Allerton [7] Michelle Bardgett [7] Emma Clark [7] Dawn Clewes [7] Cristina Cotobal Martin [3] Mary Doona [2] Jennifer A Doyle [1] Katherine Frith,[7] Helen C Hancock [7] Ailsa L Hart [8,9] Victoria Hildreth,[7] Peter M Irving [10,11] Sameena Iqbal [3] Ciara Kennedy [7] Andrew King [2] Sarah Lawrence,[7] Charlie W Lees [12,13] Robert Lees [2] Laura Letchford,[3] Trevor Liddle,[14] James O Lindsay [15,16] Rebecca H Maier [7] John C Mansfield [1,2] Julian R Marchesi [17] Naomi McGregor [7] Rebecca E McIntyre [3] Jasmin Ostermayer [3] Tolulope Osunnuyi [3] Nick Powell [17,18] Natalie J Prescott [19] Jack Satsangi [20] Shriya Sharma [7] Tara Shrestha,[7] Ally Speight [1,2] Michelle Strickland [3] James MS Wason [21] Kevin Whelan [22] Ruth Wood [7] Gregory R Young [1] Xinyue Zhang [21] Miles Parkes [6] Christopher J Stewart [1] Luke Jostins-Dean [23] Christopher A Lamb [1,2]

CJS, LJ-D and CAL are joint senior authors.

For numbered affiliations see end of article.

**Correspondence to**
Dr Christopher A Lamb;
christopher.lamb@newcastle.ac.uk

## ABSTRACT

**Introduction** Characterised by chronic inflammation of the gastrointestinal tract, inflammatory bowel disease (IBD) symptoms including diarrhoea, abdominal pain and fatigue can significantly impact patient's quality of life. Therapeutic developments in the last 20 years have revolutionised treatment. However, clinical trials and real-world data show primary non-response rates up to 40%. A significant challenge is an inability to predict which treatment will benefit individual patients.

Current understanding of IBD pathogenesis implicates complex interactions between host genetics and the gut microbiome. Most cohorts studying the gut microbiota to date have been underpowered, examined single treatments and produced heterogeneous results. Lack of cross-treatment comparisons and well-powered independent replication cohorts hampers the ability to infer real-world utility of predictive signatures.

IBD-RESPONSE will use multi-omic data to create a predictive tool for treatment response. Future patient benefit may include development of biomarker-based treatment stratification or manipulation of intestinal microbial targets. IBD-RESPONSE and downstream studies have the potential to improve quality of life, reduce patient risk and reduce expenditure on ineffective treatments.

**Methods and analysis** This prospective, multicentre, observational study will identify and validate a predictive model for response to advanced IBD therapies, incorporating gut microbiome, metabolome, single-cell transcriptome, human genome, dietary and clinical data. 1325 participants commencing advanced therapies will be recruited from ~40 UK sites. Data will be collected at baseline, week 14 and week 54. The primary outcome is week 14 clinical response. Secondary outcomes include clinical remission, loss of response in week 14 responders, corticosteroid-free response/remission, time to treatment escalation and change in patient-reported outcome measures.

**Ethics and dissemination** Ethical approval was obtained from the Wales Research Ethics Committee 5 (ref: 21/WA/0228). Recruitment is ongoing. Following study completion, results will be submitted for publication in peer-reviewed journals and presented at scientific meetings. Publications will be summarised at www.ibd-response.co.uk.

**Trial registration number** ISRCTN96296121.

⇒ IBD-RESPONSE will establish one of the largest biorepositories of longitudinal stool, blood, matched tissue and organoids with detailed linked phenotypic, diet, genetic and treatment outcome data in patients with IBD.

⇒ This study will use a standardised yet pragmatic methodology for sample collection, processing and storage, allowing multicentre participation, improving generalisation and transferability of findings to real-world clinical practice.

⇒ Remote 'at home' patient data and stool collection will allow ~40 IBD centres to contribute to recruitment, increasing power in this multi-omics study when compared with other studies in the field that generally recruit from one or a small number of centres.

⇒ While recruiting sites are limited to the UK only, an identified cross-validation cohort will increase generalisability of our findings.

## INTRODUCTION

Crohn's disease (CD) and ulcerative colitis (UC) are the principal forms of inflammatory bowel disease (IBD).[1] Characterised by symptoms including diarrhoea, rectal bleeding, abdominal pain and extraintestinal features such as fatigue, IBD can have a substantial negative impact on patient's quality of life.[2] Approximately 20% of patients with CD and 10% of patients with UC are unable to work due to their condition.[3] The global prevalence of IBD is rising. In the UK, 1 in 125 people are affected, with prevalence expected to reach 1 in 100 by 2028.[4 5] Outside of Western Europe and North America, the incidence is rising rapidly in many regions including South America, Latin America, Asia and Africa.[6 7]

The biologics era has revolutionised IBD treatment in the last 20 years. Patients and clinicians have more advanced therapies to choose from than ever before. Several biologic classes are now licensed in the UK, targeting tumour necrosis factor alpha (TNFα) (including infliximab and adalimumab), interleukin (IL)-12 and/or IL-23 cytokine pathways (ustekinumab, risankizumab and mirikizumab) or the gut-homing α4β7 integrin (vedolizumab). In addition, recently licensed small molecule therapies for UC include the Janus kinase inhibitors (JAKi) tofacitinib, filgotinib and upadacitinib, and the sphingosine-1-phosphate receptor (S1PR) modulators ozanimod and etrasimod. With several additional therapies in advanced stages of development or having completed phase III randomised clinical trials, the number of treatments available to patients is likely to increase.[8]

Current understanding of biological mechanisms driving the pathogenesis and natural history of IBD implicates complex interactions between host genetics and the gut microbiome (bacteria, viruses, fungi, archaea and phage).[9] While large clinical cohorts for human genetic discovery have led to major advances in understanding disease pathogenesis,[10] cohorts for

the study of gut microbiota have mostly been underpowered. Nonetheless, existing research intriguingly suggests utility of microbiome signatures in predicting response to therapy. In a small prospective study of 85 patients starting vedolizumab therapy, greater alpha-diversity and higher abundance of *Roseburia inulinivorans* and a Burkholderiales species at baseline were associated with therapeutic-induced remission in CD.[11] Incorporation of microbial taxonomy data alongside clinical data in a predictive model produced an area under the receiver operating characteristic curve (AUC) of 0.776 (compared with an AUC of 0.619 using clinical data only). A larger study of 232 patients receiving ustekinumab implicated *Bacteroides* and *Faecalibacterium* as predictors of treatment response.[12] Here, a predictive model of response to ustekinumab using clinical metadata produced an AUC of 0.616, rising to 0.844 when combined with baseline bacterial profile data.

Beyond IBD, further proof of concept that the gut microbiome is of prognostic importance in the context of systemically administered immune-targeted therapies is found with immune checkpoint inhibitor treatment in cancer, where experimental animal data demonstrated the beneficial impact of microbial modulation on treatment outcome.[13–15]

Metabolites derived from the gut microbiome are important intermediaries in the host-microbiome dialogue.[16] Specific classes of metabolites, such as bile acids (BA), short-chain fatty acids (SCFA) and tryptophan metabolites, may play a role in modulating disease activity and treatment responsiveness in IBD.[17] In a study of 185 (77 UC, 108 CD) patients commencing anticytokine (anti-TNFα or anti-IL-12/IL-23) or anti-integrin (vedolizumab) therapy, metabolomic and proteomic analysis of blood in addition to taxonomic and functional profiling of stool samples was conducted.[18] Among patients receiving anticytokine therapy, 120 enzymes were differentially abundant in baseline samples of remitters versus nonremitters. Single-species dominance (>50% of enzyme copies in >50% of samples explained by a single species) was observed for 8/120 enzymes. *Eggerthella lenta* was dominant for five of these eight enzymes, three of which are involved in secondary BA biosynthesis. Metabolomic analysis of baseline blood samples revealed significant enrichment of serum secondary BAs in patients achieving week 14 clinical remission. Paired baseline stool samples revealed a significant positive correlation with the abundance of 7α/β-dehydroxylation enzymes (responsible for primary to secondary BA conversion), presence of which was associated with a preferential response to anticytokine therapy. This was replicated by the authors in a small validation cohort of 46 patients initiating anti-TNFα (infliximab) therapy. In a small study of 29 patients with moderate-to-severe UC receiving vedolizumab treatment, untargeted metabolomic analysis

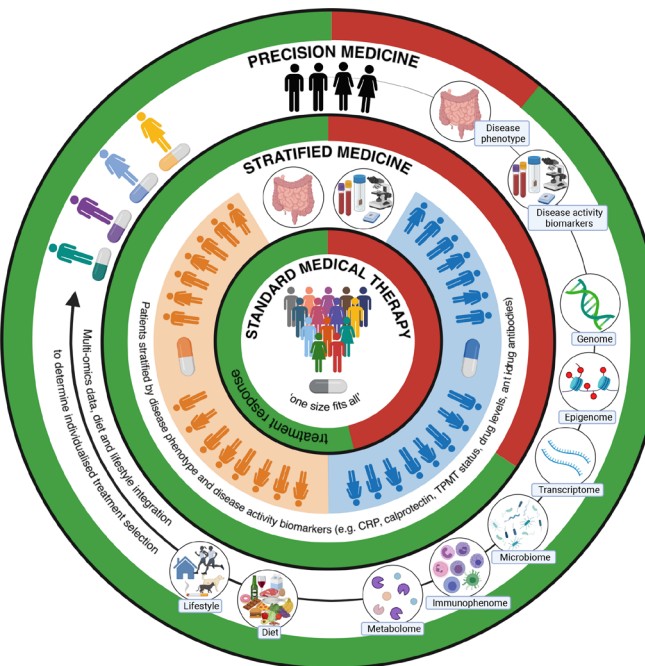

**Figure 1** Schematic illustrating the evolving approach to treatment of inflammatory bowel disease, with the aim of improving treatment outcomes through individualised precision medicine. Current treatment selection is stratified and modified based on diagnosis, disease phenotype, imaging (radiological and endoscopic) and limited clinical biomarkers, such as blood and stool markers of inflammation, drug metabolising enzyme activity, drug levels and antidrug antibodies. Precision medicine approaches integrating additional complex multi-omic data with information about environmental factors such as dietary intake, smoking and physical activity levels (the exogenous 'exposome'), may enable individualised treatment selection through predictive modelling. Precision medicine may also help to identify at-risk populations, predict disease course, reduce unnecessary patient risk and health service costs associated with ineffective pharmacological therapies and guide non-pharmacological interventions such as dietary modification (figure created with BioRender.com). CRP, C reactive protein; TPMT, thiopurine methyltransferase.

of stool showed significantly higher levels of SCFAs including butyrate in those achieving week 14 remission (defined as total Mayo score ≤2, all subscores ≤1) vs non-remitters.[19] Combining metabolite data (SCFA levels for butyrate and isobutyric acid) with microbial profile data predicted anti-integrin response with an AUC of 0.961.

While these previous studies are important first steps to using gut microbial signatures in stratified treatment algorithms, they were mostly underpowered, examined single treatments, used different sequencing technologies and produced heterogeneous non-overlapping results. The lack of cross-treatment comparisons and well-powered independent replication cohorts hampers the ability to infer real-world utility of these predictive signatures,

and to move from observations of association to causation in IBD.[20]

## METHODS AND ANALYSIS

### Study rationale

A significant challenge to effective, personalised use of biological or small molecule therapies (collectively termed 'advanced therapies') in IBD is an inability to predict which class of treatment is most likely to benefit an individual patient (figure 1). Despite increasing therapeutic options, clinical trial and real-world efficacy data show primary non-response rates of up to 40% across all therapeutic classes and in those with initial symptomatic benefit, up to 40% lose response by 1 year.[1 21–23] Consequently, complications of chronic, active inflammation including strictures, fistulae and malignancy, continue to affect a substantial number of patients and have a negative impact on patient's quality of life.[2] Up to 30% of patients with CD require surgical intervention within 10 years of diagnosis, and around 15% of patients with UC will ultimately require a colectomy.[24] Off-target side effects may also occur, including infection and malignancy.[1] With average treatment costs of £6156/year for CD and £3084/year for UC, future treatment algorithms must avoid the potential morbidity and additional cost associated with expensive treatments that do not benefit individual patients.[25]

The absolute importance of precision medicine research to identify biomarkers for treatment stratification and develop prognostic algorithms was highlighted by two recent national research prioritisation exercises incorporating responses from almost 3000 patients, their families and friends.[1 26] Validated prognostic models for treatment stratification do not exist and understanding of mechanisms controlling treatment non-response is limited. Through a multi-omic, precision medicine approach, the IBD-RESPONSE study seeks to improve selection of the right drug, for the right patient, at the right time. Other translational outputs of IBD-RESPONSE could bring into focus potential non-pharmacological approaches to treating IBD that do not necessarily involve large health economic expenditure. This could include manipulating the gut microbiome via the microbiota, through refinement of faecal microbial transplant protocols, use of prebiotics and probiotics and dietary interventions.

### Scientific objectives of IBD-RESPONSE

The primary scientific objective of IBD-RESPONSE is to identify and validate a predictive model for clinical response or failure to respond to advanced therapies in IBD after 14 weeks of therapy (the primary clinical outcome, see 'Primary clinical outcome measures' section). Modelling will incorporate gut microbiome, human genome, blood and intestinal single cell transcriptome data and detailed clinical data. Through data derived from a nested subcohort (CD-metaRE-SPONSE), predictive modelling will also include

## Box 1 IBD-RESPONSE clinical outcome measures

**Primary clinical outcome measures**
⇒ Clinical response at week 14.

**Secondary clinical outcome measures**
⇒ Clinical remission at week 14.
⇒ Clinical response at week 54.
⇒ Clinical remission at week 54.
⇒ Loss of response at week 54 in week 14 responders.
⇒ Durable corticosteroid-free response or remission at week 54 defined as receiving no corticosteroids between week 14 and week 54 assessments inclusive and not meeting criteria for loss of response.
⇒ Time to treatment escalation from baseline, defined as:
　⇒ Advanced therapy switch due to lack of efficacy/those with loss of response (does not include biosimilar switch or switch from intravenous to subcutaneous route).
　⇒ Dose intensification of drug due to lack of efficacy (does not include intensification based on therapeutic drug monitoring without flare in responders).
　⇒ Resectional intestinal surgery (does not include examination under anaesthesia procedures in patients with perianal CD).
　⇒ Induction or dose escalation of corticosteroids.
⇒ Time to treatment escalation as defined above, but disregarding dose intensification.
⇒ Time to treatment escalation as defined above, among week 14 responders.
⇒ Time to discontinuation of index drug (persistence).
⇒ Incidence of and time to potential side effects of treatment during follow-up.
⇒ Continuation of drug at week 14 and/or week 54 in those not meeting criteria for response and/or remission.
⇒ Change from baseline SF subscore at week 14 and/or 54 (both CD and UC).
⇒ Change from baseline RB subscore at week 14 and/or 54 (UC only).
⇒ Change from baseline AP subscore at week 14 and/or 54 (CD only).
⇒ Development of antidrug antibodies by week 14 or 54.
⇒ Change in CRP from baseline at week 14 or 54 (50% reduction or absolute value ≤5 mg/L deemed as clinically significant).
⇒ Change in faecal calprotectin from baseline at week 14 or 54 (50% reduction or absolute value ≤100 µg/g deemed as clinically significant).
⇒ Endoscopic remission during follow-up (Mayo endoscopic subscore ≤1 for UC or SES-CD ≤2 for CD).
⇒ Change in quality of life, physical activity, dietary intake, joint pain and fatigue as measured by study questionnaires.

AP, abdominal pain; CD, Crohn's disease; CRP, C reactive protein; RB, rectal bleeding; SES-CD, simple endoscopic score for Crohn's disease; SF, stool frequency; UC, ulcerative colitis.

## Box 2 IBD-RESPONSE exploratory scientific objectives

⇒ Test the association of microbial metabolites (metabolome) in stool or plasma, human genetics and/or single cell transcriptome data from blood or intestinal tissue with the above primary and secondary objectives.
⇒ Determine the influence of diet on the gut microbiome and treatment response in IBD, and the factors associated with dietary intake in IBD.
⇒ Explore host human genetic-gut microbiome-metabolome interactions in IBD pathogenesis and causal pathways to treatment response.
⇒ Ascertain the utility of archived endoscopy collected FFPE biopsies at predicting/imputing the gut microbiome and for inclusion in the predictive model.
⇒ Establish a longitudinal tissue, organoid and stool biobank from this well-characterised clinical cohort.

FFPE, formalin-fixed paraffin embedded; IBD, inflammatory bowel disease.

## Study design

The design of IBD-RESPONSE and the nested CD-metaRESPONSE studies are summarised in figure 2. This prospective, observational, multicentre, cohort study will recruit participants with IBD (CD, UC, IBD-unclassified (IBD-U)) who are due to commence either biologic, JAKi or S1PR modulator therapy for symptomatic, clinically active (moderate-to-severe) luminal disease. Participants do not have to be naïve to advanced therapies and may be recruited when switching within or between class of advanced therapy. Participants may be taking or planned to start concurrent thiopurines or methotrexate as combination therapy. Participation in the study will not change standard clinical care received. Detailed longitudinal clinical data will be collected alongside stool, blood and (where possible) biopsy samples, patient-reported outcome measures and dietary intake. Planned recruitment of 1325 participants will consist of approximately 762 patients with CD and 563 patients with UC (or IBD-U).

Data will be collected at baseline (prior to starting treatment), week 14 (following completion of induction therapy) and week 54. Participants will collect stool samples at each study timepoint. If a participant attends hospital within the baseline and/or week 14 study window, two blood samples (per timepoint) will be collected. Where a participant undergoes endoscopy as part of routine clinical care during the study period, up to 12 research biopsies will be collected.

Of 762 participants with CD, 300 will be consented to take part in the nested CD-metaRESPONSE subcohort. Inclusion criteria are identical to the main cohort. Clinical data collection will occur at the same timepoints as in the main cohort. Additional study components will include completion of a 4-day food diary questionnaire, capturing all food and drink consumed at the baseline and week 14 timepoints. This is in addition to the food frequency questionnaire completed by all participants. CD-metaRESPONSE participants will also be required to

detailed dietary information and blood and faecal metabolome data. The co-primary scientific objective of IBD-RESPONSE is to determine the relationship between clinical response and remission at week 14 and baseline gut microbiome.

Secondary scientific objectives of IBD-RESPONSE are to determine if there is a relationship between the microbiome at baseline or changes in the microbiome following advanced therapy with any of the secondary clinical outcomes (box 1). Further exploratory scientific objectives of IBD-RESPONSE are listed in box 2.

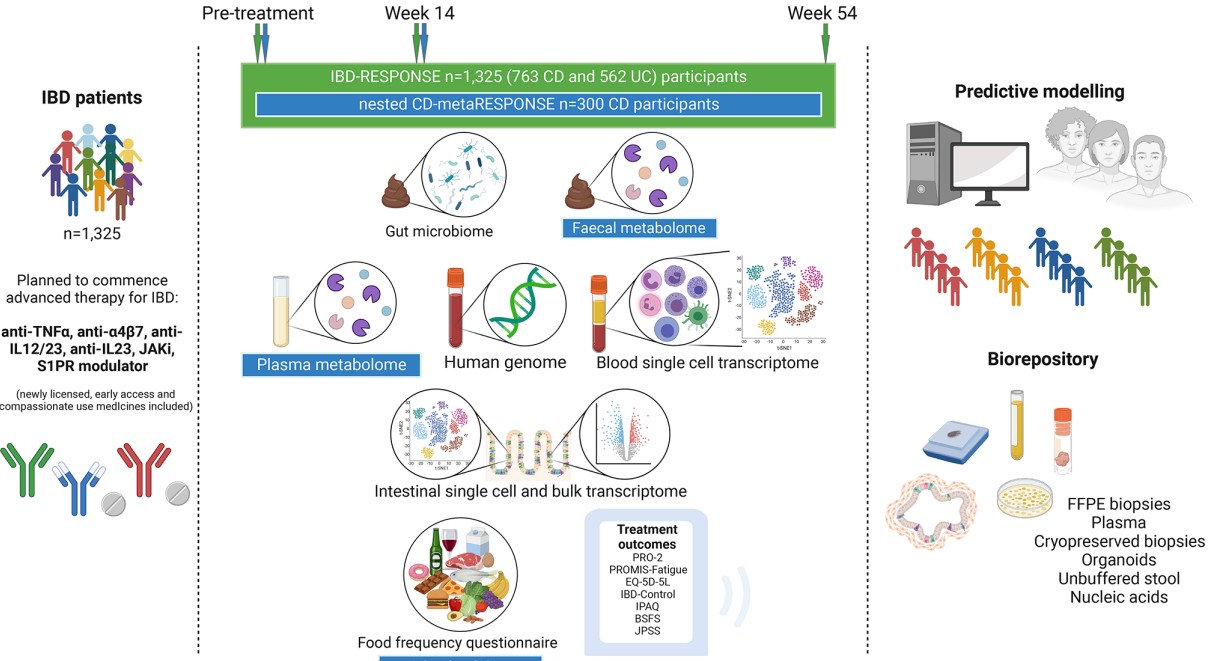

**Figure 2** Study overview schematic. 1325 participants with IBD planned to commence an advanced therapy will be recruited, including a nested subcohort of 300 patients with CD (CD-metaRESPONSE). All participants will collect two stool sample tubes at each study assessment timepoint (baseline, week 14 and week 54). CD-metaRESPONSE participants will be required to collect a third stool sample tube at baseline and week 14. If a participant attends hospital for a face-to-face appointment within the baseline and/or week 14 study assessment window, blood samples will be collected. If a participant attends hospital for a lower gastrointestinal endoscopy at any time during the study period (pre-treatment or post-treatment), biopsy samples will be collected. Participants will complete several questionnaires at each assessment timepoint. For CD-metaRESPONSE participants, additional detailed analyses will be undertaken of metabolic profiles (metabolome) in stool and matched blood plus in-depth dietary assessment (additional elements highlighted in blue boxes). Data generated will be used to perform predictive modelling. Any remaining participant samples will form a large biorepository for use in future research (figure created with BioRender.com). Anti-TNFα, antitumour necrosis factor alpha; anti-IL, anti-interleukin; BSFS, Bristol Stool Form Scale; CD, Crohn's disease; FFPE, formalin-fixed paraffin embedded; IBD, inflammatory bowel disease; IPAQ, International Physical Activity Questionnaire; JAKi, Janus kinase inhibitor; JPSS, Joint Pain and Stiffness Score; PRO-2, patient-reported outcome-2; PROMIS, Patient-Reported Outcomes Measurement Information System; S1PR, sphingosine-1-phosphate receptor; UC, ulcerative colitis.

provide two additional stool tubes (one at baseline and one at week 14) for faecal metabolome analysis. Participants will be recruited to CD-metaRESPONSE from a limited number of the participating sites. These sites will initially preferentially recruit eligible participants with CD to CD-metaRESPONSE. Once the recruitment target of 300 participants is achieved, all new participants identified with a diagnosis of CD will be recruited to the main cohort.

All participants recruited to IBD-RESPONSE will be invited to co-recruit to IBD BioResource (if not already participating). IBD BioResource is a national platform and recallable biorepository linked to the National Institute for Health and Care Research (NIHR) BioResource that is designed to expedite IBD research, currently with >45 000 participants.[27] Co-recruiting participants to IBD-RESPONSE and the IBD BioResource platform will generate a rich dataset and provide a long-term means of archiving data from IBD-RESPONSE to facilitate ongoing research and maximise downstream patient benefit. To minimise participant burden, the contact for recruitment to IBD BioResource can occur any time in the 12 months following consent to IBD-RESPONSE. If a participant ultimately decides not to participate in the IBD BioResource, they will not be withdrawn from IBD-RESPONSE.

If a participant discontinues treatment prior to week 14 or week 54 follow-up, the next timepoint assessment will be brought forward and completed as a treatment discontinuation assessment. Where a participant does not respond to the first prescribed advanced therapy and a second (or third) advanced therapy is subsequently prescribed, this discontinuation assessment will act as the baseline sample and data collection assessment for the successive advanced therapy. Follow-up samples and data collection will occur at week 14 and week 54 following commencement of each successive agent. The participant may remain in the study up to 54 weeks after commencement of a particular agent or until the end of the study period. We anticipate that up to 40% of patients will not respond to the initial prescribed therapy or will lose response by 1 year and will move on to a second (or third) advanced therapy. We therefore anticipate that

## Box 3 Eligibility criteria for IBD-RESPONSE

**Inclusion criteria**
Individuals must fulfil all the following criteria to be enrolled in the study.
⇒ Aged 16 years and over.
⇒ Diagnosis of IBD: CD, UC or IBD-U.*
⇒ Already participating or willing to be approached for participation in IBD BioResource.
⇒ Willing and able to provide informed consent.
⇒ Willing to undertake study procedures including:
  ⇒ Completion of study questionnaires
  ⇒ Collection of home stool specimens
⇒ Provision of blood and (where applicable) biopsy specimens.
⇒ Symptoms of active luminal IBD (see definition of 'Clinically active disease', table 1).
⇒ At least one biochemical, endoscopic or radiological marker of active disease within 16 weeks of study consent (see boxes 4 and 5)
⇒ Intention of clinical team to commence one of the following advanced therapies for active luminal IBD within 6 weeks of consent:△
⇒ Infliximab
⇒ Adalimumab
⇒ Vedolizumab
⇒ Ustekinumab
⇒ Risankizumab (CD only)
⇒ Mirikizumab (UC only)
⇒ Tofacitinib (UC only)
⇒ Filgotinib (UC only)
⇒ Upadacitinib
⇒ Ozanimod (UC only)
⇒ Etrasimod (UC only)

**Exclusion criteria**
Individuals meeting any of the following criteria will not be eligible to participate in the study:
⇒ Receiving oral corticosteroids for any indication where the dose is unlikely to be weaned by week 14.#
⇒ Planned bowel resection surgery within 14 weeks of commencing therapy.
⇒ Advanced therapy being commenced as rescue for ASUC.
⇒ Advanced therapy being commenced as part of a CTIMP.
⇒ Presence of an ileal pouch anal anastomosis.
⇒ Presence of a stoma.
⇒ Perianal CD in the absence of active luminal inflammation.
⇒ Antibiotics or short-term (≤4 weeks) use of probiotics within the preceding 2 weeks.†
⇒ FMT within the preceding 12 weeks or planned FMT within 14 weeks of commencing advanced therapy for IBD.‡

*Individuals with IBD-U will be managed as per the UC relevant protocol.
△Participants may be advanced therapy-naïve or -exposed. Any new biologic or small molecule drug that becomes licensed for the treatment of IBD during the planned study period will be permitted to allow study inclusion. Drugs used through Early Access to Medicines Schemes, compassionate use or expanded access schemes for unlicensed therapies are also permitted. Patients starting immunosuppressant monotherapy with a thiopurine or methotrexate are not eligible to take part. However, use of these treatments as part of combination therapy with an advanced therapy is not an exclusion to enrolment.
#Examples may include long-term oral steroids for IBD, where weaning by 14 weeks may not be possible irrespective of response to advanced

Continued

## Box 3 Continued

therapy, or concurrent diagnosis where long-term oral steroids are used, for example, polymyalgia rheumatica.
†Use of long-term (>4 weeks), stable doses of probiotics does not exclude individual participation but should be noted in the eCRF.
‡Use of antibiotics or prior FMT outside the exclusion period is permitted. Antibiotic use in the preceding 1 year and ever having received FMT will be noted in the eCRF.

ASUC, Acute severe ulcerative colitis; CD, Crohn's disease; CTIMP, Clinical Trial of an Investigational Medicinal Product; eCRF, electronic case report form; FMT, faecal microbial transplantation; IBD, inflammatory bowel disease; IBD-U, inflammatory bowel disease-unclassified; UC, ulcerative colitis.

recruitment of 1325 participants may capture approximately 1760 new treatment initiation episodes.

### Study setting

This multicentre cohort study will be conducted at ~40 study centres, based within National Health Service (NHS) Trusts across the UK. All sites must be able to accommodate the needs of the IBD-RESPONSE cohort including clinical engagement, research nurse support and facilities for assessments.

### Eligibility criteria

Individuals will be deemed eligible to enrol in the study if they fulfil all inclusion criteria and meet none of the exclusion criteria (box 3).

Participants must have at least one biochemical, endoscopic or radiological (CD only) marker of clinically active luminal disease within 16 weeks of study consent (see box 4 (CD) and box 5 (UC)). Endoscopic assessment of disease activity will be judged locally and may be assessed prospectively or retrospectively. While pregnancy may influence advance therapy selection, pregnancy is not an exclusion criterion.

### Clinical outcome measures

Key definitions related to clinical outcome measures can be found in table 1.

## Box 4 IBD-RESPONSE and CD-metaRESPONSE additional inclusion criteria: CD

**Patients with CD must also have at least one of the following documented within 16 weeks prior to consent:**
⇒ Faecal calprotectin ≥250 µg/g.
⇒ CRP ≥6 mg/L.
⇒ Any endoscopic evidence of active CD, defined as ulceration (with at least one ulcer ≥5 mm) judged locally from available clinical data (as an approximation equivalent to SES-CD of ≥4 for ileal disease or ≥6 for ileocolonic or colonic disease).
⇒ Active inflammatory disease on imaging (MRI/CT/ultrasound) judged locally from available clinical data.

CD, Crohn's disease; CRP, C reactive protein; SES-CD, simple endoscopic score for Crohn's disease.

### Primary clinical outcome measures

Clinical response at week 14 after commencing therapy (Box 1).

### Secondary clinical outcome measures

Secondary clinical outcome measures are listed in box 1.

### Sample size calculations

The sample size (n=1325 cases overall, including n=300 cases for CD-metaRESPONSE subcohort) was chosen to ensure sufficient power to answer the co-primary scientific objective (to detect associations between microbiome measures and clinical response or remission) and key exploratory scientific objectives (to detect associations between metabolites and clinical response or remission, and to detect associations between genetics and microbiome measures). Sample size calculations for predictive models require extensive assumptions about the number and effect size of associations and the correlation structure of the data. We noted predictive models built with microbiome and/or metabolite measures with high in-sample predictive accuracy (including AUC=0.78 with n=84 from Ananthakrishnan et al,[11] AUC=0.91 with n=76 from Ding et al,[28] and AUC=0.84 with n=232 from the CERTIFI study[12]), implying that n=300 individuals (from CD-metaRESPONSE) and n=1325 individuals (from IBD-RESPONSE), should be sufficient for high predictive in-sample accuracy.

### Sample size for analysis of primary objective

We took plausible effect sizes for the primary scientific objective analysis from the effect of *Bacteroides* levels in stool on ustekinumab response (d=0.66, from the CERTIFI study[12]) and the effect of antidrug immunogenicity on

remission after anti-TNFα treatment (d=0.30, from the PANTS study[21]). We calculated power for a simple two-sample t-test (using the R package pwr), assuming the expected non-response rate at 14 weeks (23.8%) and non-remission rate at 54 weeks (63.1%) from the PANTS study. We assumed a conservative Bonferroni-corrected significance threshold of 1e-5 (correcting for 5000 microbiome measures). We calculated the minimum sample size required to achieve 80% power for the two effect sizes and two outcomes (online supplemental file 1), showing that relatively low sample sizes are required to have high power to detect the larger plausible effect size (n=282 for remission and n=361 for response), but that larger sample sizes are required to have high power to detect the smaller plausible effect size (n=1331 for remission and n=1705 for response). The sample size of n=1325 chosen for our study gives a power of close to 100% for

---

### Box 5  IBD-RESPONSE additional inclusion criteria: UC and IBD-U

**Patients with UC/IBD-U must also have at least one of the following documented within 16 weeks prior to consent:**
⇒ Faecal calprotectin ≥250 µg/g.
⇒ CRP ≥6 mg/L.
⇒ Any endoscopic evidence of at least moderately active UC (of any extent including proctitis), defined as features of MCS endoscopic subscore ≥2 (marked erythema, lack of vascular pattern, friability, erosions, spontaneous bleeding or ulceration).

CRP, C reactive protein; IBD-U, inflammatory bowel disease-unclassified; MCS, Mayo Clinic Score; UC, ulcerative colitis.

---

| Table 1 | IBD-RESPONSE key clinical definitions |
|---|---|
| Clinically active disease | ► CD: unweighted PRO-2 (CD) of average daily SF subscore ≥4 and/or average daily AP subscore ≥2.<br>► UC: total PRO-2 (UC) ≥3 with RB subscore ≥1.<br>PRO-2 data will be entered by patients over 4 days (minimum 2 days PRO-2 data are permissible for PRO-2 calculation). |
| Clinical remission | Patient remains on drug and meets the following criteria:<br>► CD: unweighted PRO-2 (CD) average daily SF subscore ≤2.8 and average daily AP subscore ≤1 (and neither worse than inclusion scores at baseline).<br>► UC: PRO-2 (UC) SF subscore ≤1 with a decrease of ≥1 point(s) in SF subscore from baseline, plus RB subscore=0.<br>AND an absence of any of the following at time of assessment:<br>► Resectional bowel surgery at any time after baseline until time of current assessment.<br>► Current use of oral corticosteroids/failure to wean oral corticosteroids prescribed at baseline. |
| Clinical response | Meeting criteria for clinical remission OR:<br>► CD: unweighted PRO-2 (CD) ≥30% reduction in average daily SF subscore and/or ≥30% decrease in average daily AP subscore (and neither worse than inclusion scores at baseline).<br>► UC: total PRO-2 (UC) decrease ≥1 and ≥30% from baseline, and a decrease in RB subscore ≥1 or an absolute RB subscore of ≤1.<br>AND an absence of any of the following at time of assessment:<br>► Resectional bowel surgery at any time after baseline until time of current assessment.<br>► Current use of oral corticosteroids/failure to wean oral corticosteroids prescribed at baseline. |
| Week 14 non-response | ► Not meeting clinical response criteria AND not having stopped drug for any reason other than lack of efficacy. |
| Week 54 loss of response | ► Not meeting clinical response criteria at week 54 having met clinical response criteria at week 14 (AND not having stopped drug for any reason other than inefficacy between week 14 and week 54 assessments). |

AP, abdominal pain; CD, Crohn's disease; PRO, patient-reported outcome; RB, rectal bleeding; SF, stool frequency; UC, ulcerative colitis.

---

the larger *Bacteroides* effect size for both week 14 response and week 54 remission, and 57% and 80% for the smaller antidrug immunogenicity effect size for week 14 response and week 54 remission, respectively.

## Sample size for analysis of secondary and exploratory scientific objectives

We took plausible effect sizes for the exploratory metabolite analysis from the effects of three selected stool lipid and BA metabolites on anti-TNFα response from Ding *et al*[28]: faecal triglyceride (d=1.00), and two BA metabolites; BA1 (d=0.89) and BA3 (d=0.70). We assumed a conservative Bonferroni-corrected significance threshold of 5e-5 (correcting for 1000 metabolites). For n=300, this gives a power of 99.7%, 91% and 74% for faecal triglyceride, BA1 and BA3 for week 14 response and 100%, 99.8% and 91% for week 54 remission.

We do not anticipate that this study on its own will be well powered to detect new associations between genotype and microbiome measures in IBD. It is known from studies of healthy individuals that genetic variants that explain >3% of variation ($R^2$=0.03) in microbial abundance are rare.[29] Assuming a conservative significance threshold of <1e-11 (correcting for 1e6 independent genotypes and 5000 microbiome measures), n=1325 samples would only have 33% power to detect associations with $R^2$=0.03 (calculated using genpwr[30]). We will therefore combine our samples with a further genotype/microbiome study of IBD (PREdiCCt) to increase sample size to n=2325. This will provide us 80% power to detect genetic associations with $R^2$>0.025. In the case where such genotype/microbiome measure associations exist, and are associated with a causal biomarker for week 54 remission, a Mendelian randomisation analysis would have >80% power to demonstrate causality of this biomarker when the causal effect ORs is >2 (calculated using mRnd[31]).

## Study procedures and measures

Participants aged ≥16 years may be identified from a variety of settings such as outpatient clinics (face-to-face or virtual), flare assessments, IBD clinical nurse specialist helplines/email contact, endoscopy examinations, infusion suites, multidisciplinary team meetings and virtual biologics clinics (figure 2). Consent will be taken electronically using a Research Electronic Data Capture (REDCap) online database. Full informed e-consent will be supported by an appropriately delegated member of the study team, using a laptop, tablet or mobile device in the patient's own home, or using a hospital tablet, laptop, computer or patient mobile device during a scheduled visit to hospital as part of routine clinical care. Paper copies of the consent form will be made available for those patients unable to access e-consent. If a later decision is made not to commence an advanced therapy for IBD, the participant will not be eligible to continue in the study and must be withdrawn. Data collected up to the point of withdrawal may be used for the study. Any

samples collected will be used for research within IBD-RESPONSE or stored for future research.

All research activity will be completed by the participant either remotely or during hospital visits scheduled to deliver routine clinical care (figure 3). Participants will be asked to complete data collection after consent and before starting treatment (baseline), and at week 14 and week 54 following commencement of advanced therapy in line with routine dosing schedule visits. This will include patient questionnaires, stool samples and, where applicable, blood and biopsy specimens. Participants will be asked to complete questionnaires with data entered directly into the study specific REDCap database. Paper questionnaires will be made available for those participants without access or who express a preference to complete in paper format. Participants will be asked to complete questionnaires related to disease activity, physical activity, quality of life and diet at the three assessment timepoints (box 6).

## Participant samples

All sample collection and processing will be standardised, with full requirements detailed in a study Sample Collection Manual.

Stool samples will be collected by participants at home using stool collection kits and returned using a prepaid Royal Mail Safebox. All participants will collect two stool samples per study assessment timepoint (DNA Genotek OMNIgene•GUT tube and universal polystyrene tube). Participants recruited to the CD-metaRESPONSE subcohort will be required to collect a third stool sample at baseline and week 14 (DNA Genotek OMNImet•GUT tube). DNA extraction and metagenomic shotgun sequencing will be performed on buffered samples. Calprotectin will be measured in unbuffered stool. Remaining fresh stool and nucleic acids will be cryopreserved for use in future research.

Participants who attend a clinical appointment prior to commencing advanced therapy or within the week 14 visit window (week 10–20; week 12–16 preferred) will be asked to provide two blood samples (lithium heparin tube and EDTA tube). Blood samples will be used for single cell analysis, plasma extraction and cryopreservation. Any remaining blood samples or derivatives will be stored at Newcastle Biobank for use in future research. Participants will not be asked to attend hospital specifically for blood sample collection. If participants are not scheduled to attend hospital face-to-face within either study assessment window, blood samples will not be taken.

If a participant has a lower gastrointestinal endoscopy as part of planned care during study participation, up to 12 research biopsies (one set of 6 biopsies to be collected from the colon in all participants and a further set of 6 biopsies from the ileum in participants with CD) will be taken. Where a participant meets the study eligibility criteria and a disease assessment endoscopy is planned prior to starting therapy, consent should be received ahead of their planned endoscopy to enable the collection of

**Potential participant identified**

**Where:** Face-to-face clinics, telephone consultations, endoscopy lists, infusion suites, MDT meetings, virtual biologics clinics, IBD helpline
**Who:** ≥16 years old; symptomatic, active luminal IBD; no minimum disease duration
**Advanced therapy being commenced:** Licensed biologic, JAKi or S1PR modulator (infliximab, adalimumab, vedolizumab, ustekinumab, risankizumab (CD), mirikizumab (UC), tofacitinib (UC), filgotinib (UC), upadacitinib, ozanimod (UC), etrasimod (UC)) or early access/compassionate use therapy
**Screening tips:**
- UC symptoms: increased stool frequency relative to baseline AND at least one episode of recent rectal bleeding within e.g. last 3-5 days
- CD symptoms of active flare: presence of liquid or very soft stools AND/OR presence of abdominal pain
- Active disease confirmed by clinical test in 16 weeks prior to consent: faecal calprotectin, CRP, endoscopy or (CD only) imaging
- Participants do not need to be naïve to advanced therapies e.g. can previously have received an advanced therapy
- Participants switching between advanced therapies do not need a minimum washout period
- Participants can be receiving or about to start concomitant thiopurine or methotrexate alongside planned advanced therapy
- Participants can be receiving or starting steroids at baseline provided likely to have fully weaned by week 14

**Eligibility confirmed**

**Study information and informed consent given**

- Participant Information Sheet (PIS) provided (paper or electronic format)
- Potential participant questions answered by local study team
- REDCap access given to participant to complete informed study consent

**Consent verified**

Stool sample collection kit given to/posted to participant once valid consent obtained and verified by local study team

**Baseline assessment**

**Ideal: Week 0**
Target: Up to 6 weeks prior to commencing advanced therapy
Permitted: Assessment >6 weeks prior to commencing advanced therapy is permitted, e.g. if delayed due to infusion unit capacity

**All participants to complete after consent and before starting advanced therapy:**
1. Participant questionnaires completed on REDCap
2. Stool sample tubes (n=2) collected and returned using prepaid Royal Mail Safebox™
3. If attending hospital for a face-to-face clinical encounter within assessment window, two tubes (total 20mls) blood collected

**CD-metaRESPONSE subcohort (in addition to above requirements):**
1. Third stool sample tube collected (provided in home stool sample collection kit)
2. Prospective 4-day food diary completed

Participants unable or unwilling to complete the PRO-2 questionnaire, not meeting the PRO-2 threshold for clinically active disease, missing stool samples or who do not start treatment with an advanced therapy will be withdrawn from the study

**Eligibility re-confirmed**

Baseline 4 day PRO-2 (minimum 2 days data) assessed to confirm inclusion criteria for clinically active disease met

**Biologic, JAKi or S1PR modulator commenced**    **WEEK 0**

**Week 14 assessment**

**Ideal: Week 14**
Target: Week 12 - 16
Permitted: Week 10 - 20

As per baseline assessment (including additional requirements for CD-metaRESPONSE)

If a participant stops treatment, the next planned study assessment should be brought forward and completed as a treatment discontinuation assessment. Where an alternative advanced therapy is then commenced, this discontinuation assessment will serve as the baseline assessment for the next therapy. Further assessments should then be completed at week 14 and 54 after commencing a new advanced therapy as per the study protocol

**Week 54 assessment**

**Ideal: Week 54**
Permitted: Week 48 - 60

**All participants:**
1. Participant questionnaires completed on REDCap
2. Stool sample tubes (n=2) collected and returned using prepaid Royal Mail Safebox™

In addition to the above assessments, if a participant attends hospital for a planned lower gastrointestinal endoscopy at any time during the study period following consent, the following samples/data will be collected: up to 12 research biopsies, patient-reported 1-day PRO-2 score, endoscopic assessment of disease activity data (SES-CD/MCS endoscopic subscore)

IBD RESPONSE    CD RESPONSE

**Figure 3** Flow chart providing overview of study events. Crohn's disease (CD), inflammatory bowel disease (IBD), Janus kinase inhibitor (JAKi), Mayo Clinic Score (MCS), multidisciplinary team (MDT), patient-reported outcome-2 (PRO-2),Research Electronic Data Capture (REDCap), simple endoscopic score for Crohn's disease (SES-CD), sphingosine-1-phosphate receptor (S1PR), ulcerative colitis (UC).

**Box 6    Summary of patient questionnaires completed throughout study period**

**Completed by all participants at baseline, week 14 and week 54**
**Patient-reported outcome-2 (Crohn's disease (CD) or ulcerative colitis version depending on diagnosis)**
Patient-reported outcome-2 is a validated questionnaire measuring patient-reported outcomes including stool frequency, abdominal pain and rectal bleeding.[33 34]

**Bristol Stool Form Scale**
The Bristol Stool Form Scale is a 7-point scale that helps describe stool shape and consistency and assess bowel patterns and habits.[35]

**IBD-Control**
The IBD-Control questionnaire comprises 13 items plus a visual analogue scale ranging from 0 to 100.[36] The questionnaire measures patient-related outcome of their disease state during the past 2 weeks.

**Patient-Reported Outcomes Measurement Information System-Fatigue 8a Short Form**
The Patient-Reported Outcomes Measurement Information System are validated questionnaires that help evaluate patient's quality of life.[37]

**EQ-5D-5L**
The EQ-5D-5L is a quality-of-life questionnaire and is a widely used generic patient-reported outcome measures incorporating five domains: (1) mobility, (2) self-care, (3) usual activities, (4) pain/discomfort, (5) anxiety/depression.[38 39] Scores for each domain are combined to describe the patient's state of health.

**International Physical Activity Questionnaire**
International Physical Activity Questionnaire (IPAQ) is a commonly used self-reported questionnaire to estimate physical activity and sedentary behaviours for adults across a range of socio-economic settings.[40] The IPAQ measures the type of physical activities people do as part of their everyday lives.

**IBD-RESPONSE Joint Pain and Stiffness Score**
The Ankylosing Spondylitis Disease Activity Score (ASDAS) is a patient-reported questionnaire which quantifies clinical disease activity in ankylosing spondylitis and combines five disease activity variables (four 10-point Likert scale patient symptom responses and a C reactive protein measurement), to produce a single score.[41–43] We believe joint pain to be an under-recognised symptom in active inflammatory bowel disease (IBD) which may change in response to therapy as inflammation resolves/fails to resolve and so we have modified the ASDAS to assess joint pain, swelling and stiffness in all IBD-RESPONSE patients irrespective of whether they have a rheumatological diagnosis. We have called this modified score the IBD-RESPONSE Joint Pain and Stiffness Score.

**Food frequency questionnaire**
The food frequency questionnaire (FFQ) will be completed by participants directly into the study-specific REDCap database and can also be completed on paper. The FFQ requests information on 175 food items, their typical portion size and frequency of consumption and has been extensively validated for measuring nutrient intakes in adults.[44] Data are converted to nutrient intake using the Composition of Foods Integrated Dataset, as well as diet quality indices and other food components (eg, polyphenols) and food categorisation (eg, ultra-processed foods). The FFQ data will initially be analysed at the Centre for Healthcare Randomised Trials, University of Aberdeen and other collaborating institutions inlcuding the Department of Nutritional Sciences, King's College London.
**Completed by CD-metaRESPONSE participants at baseline and week 14 only**
**4-day food diar**y

Continued

**Box 6    Continued**

The 4-day food diary measures current food intake. It will comprehensively and prospectively measure all intake allowing calculation of energy and nutrient intake, dietary indices (eg, diet diversity and Mediterranean diet), intake of ultra-processed foods, prebiotic and emulsifier intake.

research biopsies. Biopsy samples will be used for single cell sequencing and organoid generation. 16S rRNA gene sequencing of both formalin-fixed paraffin embedded and fresh tissue will also be undertaken. Any remaining biopsy samples will be stored for use in future research.

## Statistical analysis

The analysis approach for our primary objective will be to test for the association between features of the patient microbiome at baseline and primary clinical response or remission to treatment at 14 weeks. Features will include alpha diversity, abundance of bacterial taxa (including species, genus and phyla) as well as the abundance of genes within various microbial pathways (eg, using Kyoto Encyclopedia of Genes and Genomes (KEGG) pathways, MetaCyc metabolic pathways and gene families). Association testing will be carried out using negative binomial regression, controlling for total sequence depth and predefined technical and clinical confounders and significance will be determined using Benjamini-Hochberg multiple testing correction to ensure a false discovery rate of <5%.

For the predictive modelling aim, we will use a random forest classifier to predict primary response to treatment at 14 weeks using microbiome, host genetic and clinical features at baseline. Model parameters will be tuned and accuracy assessed using nested cross-validation. This full model will be compared with a clinical-variables-only model, with model performance quantified by area under the receiver operator curve, as well as the sensitivity, specificity and positive and negative predictive power. A further model will be fitted using the same approach including metabolomic and dietary data on the CD-metaRESPONSE subset. Our primary predictive measure will be reported for a random forest classifier, but a further sensitivity analysis will be carried out by fitting alternative prediction models to test whether this has a strong effect on the predictive accuracy, using both simpler models (including logistic regression with a Least Absolute Shrinkage and Selection Operator (LASSO) penalty) and other more advanced methods (such as neural networks and support vector machines). The results of the predictive models will be reported in future publications according to the Transparent Reporting of a multivariable prediction model for Individual Prognosis or Diagnosis guidelines.[32]

Secondary and exploratory objective analyses will use the same general analysis approach as described above. For time-dependent events, such as treatment escalation

due to loss of response, a Cox proportional hazards regression will be used to assess the impact of microbiome features on time to event, with patients censored at 54 weeks, last recorded (if lost to follow-up) or date of withdrawal (if withdrawn). The dietary data will be analysed to test the association between primary response and measures of specific nutrients (such as dietary fibre) and dietary indices will be used to assess adherence to certain recommended diets (such as achievement of food-based dietary guidelines or a 'Mediterranean diet'). Host genotype data will be used to test for associations between primary response and generate polygenic risk scores of susceptibility to CD and UC, as well as prespecified variants associated with response to therapy (including HLA-DQA1*05). In all these individual analyses, Benjamini-Hochberg will be applied to control the false discovery rate at 5%.

Loss to follow-up and missing data will be handled during the analyses in different ways depending on the specific question being addressed. For the primary analysis at 14 weeks, individuals who are lost to follow-up or withdraw from the study before 14 weeks will be removed from the analysis, although we will also carry out a robustness analysis where we include individuals lost to follow-up as non-responders to ensure the results are robust to this choice. Secondary and exploratory analyses at specific timepoints will be treated in the same fashion. For time-dependent events, analysed using survival models, individuals who are lost to follow-up or withdraw will be treated as censored at this timepoint (the point of withdrawal for withdrawn participants, and the last point of contact for patients lost to follow-up), although we will also carry out sensitivity analyses where these are instead treated as adverse outcomes where appropriate (eg, treating withdrawals as adverse events). For missing data, standard quality control criteria for microbiome, metabolome and genetic data will be used to remove variables with excessive missing data (as well as other markers of poor data quality). Where data are missing for microbiome or other experimental assays for specific individuals after quality control, only participants with non-missing data for this variable will be analysed. When constructing and validating predictive models, individuals with missing data for the predictive variables being tested, or that have withdrawn or been lost to follow-up before the assessment time, will be excluded from model building and testing. Statistical analysis will be carried out in R.

## Replication

Scaling up microbiome discoveries and providing validation of results is needed to benefit patients. We will validate our predictive model using an appropriate, already assembled microbiome validation cohort. To generate this replication cohort, we will use banked stool DNA from the Prognostic effect of Environmental factors in Crohn's and Colitis Study (PREdiCCt), led by Professor Charlie Lees. PREdiCCt is a prospective observational study of participants with IBD in clinical remission, designed to identify whether baseline factors (including genetics, dietary habits and gut microbiota) predict subsequent disease flare. We will perform metagenomic sequencing of 1000 stool samples from patients who experienced a disease flare during the PREdiCCt study. We expect approximately 40% of PREdiCCt patients to experience a disease flare requiring commencement of biologics. As these patients all have baseline (clinical remission) stool microbiome samples in storage, they provide a well-matched and cost-efficient set of samples for replicating IBD-RESPONSE results.

## Potential future benefit to patients

IBD-RESPONSE will provide timely and important information regarding associations between the gut microbiome and responsiveness to treatment in IBD. It will likely highlight potential mechanisms through which the microbiota may drive inflammation. We hope that findings from IBD-RESPONSE will lead to new personalised avenues for IBD treatment through discovery and validation of predictive tools that may be incorporated directly into clinical practice or further tested in stratified clinical trials. This could lead to the development of experimental techniques to modify gut microbes; for example, donor selection for faecal microbial transplantation, identification of single or multiple strains of microbes or use of antimicrobials, phage or microbial metabolites that may be used to induce a more 'treatment-responsive' microbiome.

## Patient and public involvement statement

IBD-RESPONSE was informed by two national patient research prioritisation exercises in IBD care led by members of our team and involving feedback from 3000 people living with IBD, their family and friends.[1 26] These identify aspects of precision medicine, microbiome and diet in IBD as of high importance. The initial IBD-RESPONSE grant proposal to the Medical Research Council was reviewed by the NIHR Research Design Service North East and North Cumbria Patient and Public Involvement Panel and was presented and discussed at the 2020 Crohn's & Colitis UK Patient and Public Involvement in Research Day. IBD-RESPONSE has been supported by two patient representatives in the Study Oversight Committee (SOC) since inception. The study team have engaged with patient members of the SOC to ensure all patient facing documents including the participant information sheet, consent forms, stool collection guidance for patients and study questionnaires have undergone review. Patients will also be involved in dissemination activities relating to outputs from this research.

## ETHICS AND DISSEMINATION

Ethical approval for the study was obtained from the Wales Research Ethics Committee 5 (reference 21/WA/0228). Recruitment to IBD-RESPONSE began in February 2022 and is currently ongoing at sites around the UK.

In line with the Newcastle University and The Newcastle upon Tyne Hospitals NHS Foundation Trust research data policy, datasets will be kept for at least 5 years after the date they were last accessed. Metadata linked to genomic and metagenomic datasets will include anonymised clinical information. Examples include diagnosis (UC/CD/IBD-U), disease location, disease behaviour, complications, extraintestinal manifestations, comorbidities, family history, smoking history, surgical interventions and outcomes from prior drug therapies.

Raw data files in the original format (eg, fastq) and the accompanying anonymised phenotypic data will be uploaded to a public repository, for example, the NCBI database of Genotypes and Phenotypes at https://www.ncbi.nlm.nih.gov/gap/.

As part of CD-metaRESPONSE, microbial sequence and faecal/serum metabolomic data will be integrated with single-cell RNA-sequencing, human genomics and clinical outcome data; the whole dataset will be made available to other investigators and will be archived long term within the IBD BioResource to facilitate downstream research.

The Chief Investigator, Study Management Group, Sponsor, Funders and research team members are committed to ensure that the research findings are shared. Findings will be written up and submitted to a peer-reviewed scientific journal. Findings will be presented by the study team at national and international conferences, for example, the British Society of Gastroenterology annual meeting, the European Crohn's and Colitis meeting and Digestive Diseases Week. The study team will prepare a lay summary of the study findings for dissemination to the study participants and members of the national patient group, Crohn's & Colitis UK. Following study completion, results will be submitted for publication in peer-reviewed journals and presented at national and international scientific meetings.

## Author affiliations

[1]Translational & Clinical Research Institute, Faculty of Medical Sciences, Newcastle University, Newcastle upon Tyne, UK
[2]Department of Gastroenterology, Newcastle upon Tyne Hospitals NHS Foundation Trust, Newcastle upon Tyne, UK
[3]Wellcome Sanger Institute, Wellcome Genome Campus, Hinxton, UK
[4]Department of Gastroenterology, Royal Devon University Healthcare NHS Foundation Trust, Exeter, UK
[5]Exeter Inflammatory Bowel Disease and Pharmacogenetics Research Group, University of Exeter, Exeter, UK
[6]Department of Gastroenterology, Cambridge University Hospitals NHS Foundation Trust, Cambridge, UK
[7]Newcastle Clinical Trials Unit, Newcastle University, Newcastle upon Tyne, UK
[8]Department of Gastroenterology, St Mark's Hospital and Academic Institute, London, UK
[9]Department of Surgery and Cancer, Imperial College London, London, UK
[10]Department of Gastroenterology, Guy's and St Thomas' NHS Foundation Trust, London, UK
[11]School of Immunology & Microbial Sciences, King's College London, London, UK
[12]Institute of Genetics & Molecular Medicine, University of Edinburgh, Edinburgh, UK
[13]Edinburgh IBD Unit, Western General Hospital, NHS Lothian, Edinburgh, UK
[14]Research Informatics Team, Clinical Research, Newcastle upon Tyne Hospitals NHS Foundation Trust, Newcastle upon Tyne, UK
[15]Department of Gastroenterology, Barts Health NHS Trust, London, UK
[16]Centre for Immunobiology, Blizard Institute, Barts and the London School of Medicine, Queen Mary University of London, London, UK
[17]Division of Digestive Diseases, Department of Metabolism, Digestion and Reproduction, St Mary's Hospital, Imperial College London, London, UK
[18]Department of Gastroenterology, Imperial College Healthcare NHS Trust, London, UK
[19]Department of Medical and Molecular Genetics, Guy's Hospital, King's College London, London, UK
[20]Nuffield Department of Medicine, University of Oxford, Oxford, UK
[21]Population Health Sciences Institute, Faculty of Medical Sciences, Newcastle University, Newcastle upon Tyne, UK
[22]Department of Nutritional Sciences, King's College London, London, UK
[23]Kennedy Institute of Rheumatology, University of Oxford, Oxford, UK

**X** Nicola J Wyatt @wyatt_nic, Carl A Anderson @anderson_carl, Nicholas A Kennedy @DrNickKennedy, Tim Raine @IBD_MB, Tariq Ahmad @tariqahmadIBD, Dean Allerton @Dean_Allerton, Dawn Clewes @DawnClewes, Ailsa L Hart @ DrAilsaHart, Charlie W Lees @charlie_lees, Robert Lees @bob_lees92, Julian R Marchesi @gut_health, Rebecca E McIntyre @mcinty_re, Nick Powell @ NickPowellLab, Natalie J Prescott @natter5, James MS Wason @JMSWason, Kevin Whelan @ProfWhelan, Gregory R Young @mashedbanana, Christopher J Stewart @CJStewart7, Luke Jostins-Dean @lukejostins and Christopher A Lamb @ DrChrisLamb

**Acknowledgements** We are grateful for support from the Newcastle Clinical Trials Unit, the National Phenome Centre, Imperial College London and the NIHR Biomedical Research Centres from Newcastle, Imperial and Cambridge.

**Contributors** All authors contributed to clinical protocol and/or laboratory standard operating procedure design, development and operationalisation. CAL is the Chief Investigator of the programme. The grants to fund IBD-RESPONSE and CD-metaRESPONSE were conceptualised and written by CAL, LJ-D, CJS, MP, CAA, NAK, TR, TA, ALH, HCH, CWL, JCM, JRM, NP, NJP, AS and JS. Clinical study design, operationalisation, data acquisition and analytical plans were further developed by these authors plus DA, MB, EC, DC, MD, KF, VH, AK, CK, PI, JOL, RL, SL, TL, NMcG, RHM, SS, TS, HW, JW, KW, NJW, RW, GRY and XZ with critical review from all authors. Laboratory standard operating procedures were developed by CAL, CJS, HW, REM, LJ-D, CK, NAK, CAA, MB, EC, DC, JAD, MD, SI, JOL, LL, CCM, JRM, J0, TO, MP, MS, NP, NJP, TR, MS, SS and NJW. Manuscript drafting was led by NJW and CAL with subsequent critical review and revision by all authors.

**Funding** The IBD-RESPONSE cohort is supported by a grant from the Medical Research Council (funder reference MR/T032162/1) and the CD-metaRESPONSE cohort by The Leona M. and Harry B. Helmsley Charitable Trust (funder reference 2002-04255). Single-cell RNA-sequencing in IBD-RESPONSE and CD-metaRESPONSE is supported by a grant from the Helmsley Charitable Trust (funder reference 2304-05972). CJS is supported by a Sir Henry Dale Fellowship jointly funded by the Wellcome Trust and the Royal Society (grant number 221745/Z/20/Z) and the 2021 Lister Institute Prize Fellow Award. NJW is supported by the NIHR Academic Clinical Fellowship (ACF) programme. LJ-D is supported by a Sir Henry Dale Fellowship jointly funded by the Wellcome Trust and the Royal Society (grant number 208750/Z/17/Z) and the Kennedy Trust for Rheumatology Research. JMSW is funded by a NIHR Research Professorship (NIHR301614). XZ is funded by a NIHR Predoctoral Fellowship (NIHR302014).

**Disclaimer** The views expressed are those of the authors and not necessarily those of our funders, the NIHR or the Department of Health and Social Care.

**Competing interests** TA reports personal grants from F. Hoffmann-La Roche, Biogen, AbbVie, Janssen, Celltrion, Galapagos, Immunodiagnostik and Takeda, outside the submitted work; personal fees for educational development/delivery from Pfizer, payment or honoraria for lectures, presentations, speakers bureaus, manuscript writing or educational events from Pfizer, Takeda and F. Hoffman-La Roche; support for attending meetings from Celltrion, Tillotts and Pfizer. CAA reports grants from the Wellcome Sanger Institute Quinquennial Review 2021–2026, Crohn's and Colitis Foundation (USA), the Medical Research Council, Open Targets UK and the Helmsley Charitable Trust; consulting fees from BridgeBio, Genomics and Brigham & Women's Hospital Boston; payment or honoraria for lectures, presentations, speakers bureaus, manuscript writing or educational events from GlaxoSmithKline; support for attending meetings and/or travel membership from the Wellcome Sanger Institute Quinquennial Review 2021–2026; (Chair) of the Board of Trustees for the Sanger Prize; other interests as Director of Anderson Genomics Consultancy. MB reports partial personal salary funding from the Medical Research Council. ALH reports personal consulting fees from AbbVie, BMS, Celltrion, Falk, Galapagos, Janssen, Pfizer, Takeda and Roche; payment or honoraria for lectures, presentations, speakers bureaus, manuscript writing or educational events

from BMS, Celltrion, Falk, Galapagos, Janssen, Pfizer, Takeda, Roche and AbbVie; support for attending meetings and/or travel from BMS, Celltrion, Falk, Galapagos, Janssen, Pfizer, Takeda, Roche and AbbVie. PI reports personal grants from Celltrion, Galapagos and Pfizer, outside the submitted work; personal consulting fees from AbbVie, Takeda, BMS, Janssen, Arena, Pfizer, Galapagos, Lilly, Boehringer-Ingelheim and Celgene; payment or honoraria for lectures, presentations, speakers bureaus, manuscript writing or educational events from AbbVie, Takeda, Janssen, Lilly, BMS, Pfizer and Galapagos; support for attending meetings and/or travel from AbbVie and Tillotts. LJ-D reports grants from the Wellcome Trust, the Royal Society, the Kennedy Trust for Rheumatology Research, the Helmsley Charitable Trust and the Medical Research Council; grants from Novartis Pharmaceutical, outside the submitted work; consulting fees from Nightingale Health and Genomics. CK reports partial salary funding from the Medical Research Council. NAK reports grants from AbbVie, Biogen, Celltrion, Galapagos and Immunodiagnostik; consulting fees from AbbVie, Bristol-Meyers Squibb and Dr Falk; payment or honoraria for lectures, presentations, speakers bureaus, manuscript writing or educational events from AbbVie, Dr Falk, Tillotts, Galapagos and Takeda; support for attending meetings and/or travel from Tillotts; participation (Chair) on the Board of the British Society of Gastroenterology IBD Clinical Research Group. CAL reports grants from and/or consultancy for Janssen, Takeda, AbbVie, AstraZeneca, Eli Lilly, Orion, Pfizer, Roche, Sanofi Aventis, UCB, Biogen, GSK, Bristol-Myers Squibb and Genentech; payment or honoraria for lectures, presentations, speakers bureaus, manuscript writing or educational events from Ferring, Takeda, Janssen, Nordic Pharma and Dr Falk; participation (Secretary) on the British Society of Gastroenterology IBD Section; participation on the Steering Committee of IBD UK. CWL reports grants from UKRI Future Leaders Fellowship; personal consulting fees from AbbVie, Pfizer, Janssen, Takeda, Galapagos, Fresnius Kabi, Novartis/Sandoz, BMS and Celltrion; payment or honoraria for lectures, presentations, speakers bureaus, manuscript writing or educational events from AbbVie, Pfizer, Janssen, Takeda, Galapagos, Fresnius Kabi, Novartis/Sandoz, BMS, Ferring, Dr Falk and Celltrion. JOL reports grants from AbbVie, and Gilead; personal consulting fees from Allergan, AbbVie, Bristol-Myers Squibb, Celgene, Cornerstones US, Galapagos, Gilead, GSK, Lilly, MSD UK, Shire UK, Shire International, Ferring UK, Ferring International, Celltrion, Takeda, Pfizer and Janssen; payment or honoraria for lectures, presentations, speakers bureaus, manuscript writing or educational events from AbbVie, Bristol-Myers Squibb, Cornerstones US, Galapagos, Ferring UK, Ferring International, Celltrion, Takeda, Pfizer and Janssen; support to attend meetings and/or travel from AbbVie and Janssen. RHM reports that she is an independent membership on the Trial Steering Committee for the National Institute for Health and Care Research funded ALLEGRO trial. JRM reports personal consulting fees from EnteroBiotix and Cultech; patent held (without financial gain) on *Clostridium difficile* therapy (WO2019197836A1), participation (Chair) on the IDMC Board. NMcG reports partial (10%) salary funding from the Medical Research Council. REM reports personal salary funding from the Wellcome Sanger Institute. JO reports stock held in Novartis. MP reports grants from Pfizer and Gilead; personal consulting fees from Galapagos; payment or honoraria for lectures, presentations, speakers bureaus, manuscript writing or educational events from Janssen. NP reports grants from Bristol-Myers Squibb, Takeda and Pfizer; consulting fees from AbbVie, Allergan, AstraZeneca, Bristol-Myers Squibb, Celgene, Celltrion, Galapagos, GSK, Takeda and Vifor; payment or honoraria for lectures, presentations, speakers bureaus, manuscript writing or educational events from AbbVie, Bristol-Myers Squibb, Ferring, Galapagos, Janssen, Roche, Pfizer, Takeda and Tillotts; support for attending meetings and/or travel from AbbVie, Allergan, Celltrion, Janssen and Takeda; participation on a data safety monitoring board or advisory board for AbbVie, Allergan, AstraZeneca, Bristol-Myers Squibb, Celgene, Celltrion, Galapagos, GSK, Takeda and Vifor. TR reports personal grants from AbbVie; personal consulting fees from AbbVie, Arena, Aslan, AstraZeneca, Boehringer-Ingelheim, BMS, Celgene, Ferring, Galapagos, Gilead, GSK, Heptares, LabGenius, Janssen, Mylan, MSD, Novartis, Pfizer, Roche, Sandoz, Takeda, UCB and XAP therapeutics; participation on the board of UCB, membership (Chair) of the ECCO Guidelines Committee, membership of the UEG Scientific Committee. JS reports grants from Crohn's and Colitis UK, the Helmsley Charitable Trust, ECCO, the European Commission, CCFA and Action Medical Research; payment or honoraria for lectures, presentations, speakers bureaus, manuscript writing or educational events from Roche; participation on a Data Safety Monitoring Board or Advisory Board for the MODULATE trial and the TRIBUTE trial; leadership or fiduciary role as the Director of the Royal College of Physicians IBD Registry, and Governing Body Fellow at Green Templeton College. AS reports personal consulting fees from GSK; payment or honoraria for lectures, presentations, speakers bureaus, manuscript writing or educational events from Falk, and AbbVie; payment of conference fees to attend the British Society of Gastroenterology Annual Conference 2022 from Celltrion; participation on a Data Safety Monitoring Board or Advisory Board for the IBD-RESPONSE study (unpaid), and AbbVie; participation on the British Society of Gastroenterology IBD Section Committee. CJS reports personal consultancy fees from Astarte Medical; payment or honoraria for lectures, presentations, speakers bureaus, manuscript writing or educational events from Nestle Nutrition Institute. JW reports grants from Intercept; consulting fees from Worg and UCB; payment or honoraria for lectures, presentations, speakers bureaus, manuscript writing or educational events from Janssen; participation on a Data Safety Monitoring Board or Advisory Board for Roche. KW reports grants from the Helmsley Charitable Trust, Crohn's and Colitis UK, Almond Board of California, Danone, International Dried Fruit and Nut Council, Medical Research Council, National Institute for Health and care Research; royalty or license payments for Volatile organic compounds in the diagnosis and management of irritable bowel syndrome, and Wiley BDA Advances in Nutrition & Dietetics book series; personal consulting fees from Danone; payment or honoraria for lectures, presentations, speakers bureaus, manuscript writing or educational events from Janssen; support for attending meetings and/or travel from Yakult; participation on a Data Safety Monitoring Board or Advisory Board for the MODULATE trial (unpaid).

**Patient and public involvement** Patients and/or the public were involved in the design, or conduct, or reporting, or dissemination plans of this research. Refer to the 'Methods' section for further details.

**Patient consent for publication** Not applicable.

**Provenance and peer review** Not commissioned; externally peer reviewed.

**ORCID iDs**
Nicola J Wyatt http://orcid.org/0000-0001-5050-327X
Hannah Watson http://orcid.org/0000-0002-7965-938X
Carl A Anderson http://orcid.org/0000-0003-1719-7009
Nicholas A Kennedy http://orcid.org/0000-0003-4368-1961
Tim Raine http://orcid.org/0000-0002-5855-9873
Tariq Ahmad http://orcid.org/0000-0002-6058-5528
Dean Allerton http://orcid.org/0000-0003-0507-4343
Michelle Bardgett http://orcid.org/0000-0003-1393-2010
Emma Clark http://orcid.org/0000-0003-0065-1463
Dawn Clewes http://orcid.org/0000-0001-9600-859X
Cristina Cotobal Martin http://orcid.org/0000-0002-5877-2228
Mary Doona http://orcid.org/0000-0001-9533-3745
Jennifer A Doyle http://orcid.org/0000-0001-7042-3269
Helen C Hancock http://orcid.org/0000-0002-1494-8551
Ailsa L Hart http://orcid.org/0000-0002-7141-6076
Peter M Irving http://orcid.org/0000-0003-0972-8148
Sameena Iqbal http://orcid.org/0000-0003-0926-593X
Ciara Kennedy http://orcid.org/0000-0003-2987-0977
Andrew King http://orcid.org/0000-0002-5002-4751
Charlie W Lees http://orcid.org/0000-0002-0732-8215
Robert Lees http://orcid.org/0009-0003-7439-1681
James O Lindsay http://orcid.org/0000-0003-3353-9590
Rebecca H Maier http://orcid.org/0000-0002-7350-3288
John C Mansfield http://orcid.org/0000-0003-2490-7750
Julian R Marchesi http://orcid.org/0000-0002-7994-5239
Naomi McGregor http://orcid.org/0000-0002-5961-124X
Rebecca E McIntyre http://orcid.org/0000-0001-5291-1533
Jasmin Ostermayer http://orcid.org/0000-0001-6461-398X
Tolulope Osunnuyi http://orcid.org/0000-0002-0482-917X
Nick Powell http://orcid.org/0000-0003-3231-6950
Natalie J Prescott http://orcid.org/0000-0002-5901-7371
Jack Satsangi http://orcid.org/0000-0002-6357-9684

Shriya Sharma http://orcid.org/0000-0001-5237-757X
Ally Speight http://orcid.org/0000-0003-3184-9181
Michelle Strickland http://orcid.org/0000-0001-8053-400X
James MS Wason http://orcid.org/0000-0002-4691-126X
Kevin Whelan http://orcid.org/0000-0001-5414-2950
Ruth Wood http://orcid.org/0000-0002-8296-1774
Gregory R Young http://orcid.org/0000-0001-5342-1421
Xinyue Zhang http://orcid.org/0000-0002-5321-4235
Miles Parkes http://orcid.org/0000-0002-6467-0631
Christopher J Stewart http://orcid.org/0000-0002-6033-338X
Luke Jostins-Dean http://orcid.org/0000-0002-2475-3969
Christopher A Lamb http://orcid.org/0000-0002-7271-4956

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
