## [Reviewer comments · BMJ Open]

ARTICLE DETAILS

TITLE (PROVISIONAL)	Defining predictors of responsiveness to advanced therapies in Crohn's disease and ulcerative colitis: Protocol for the IBD-RESPONSE and nested CD-metaRESPONSE prospective, multicentre, observational cohort study in precision medicine
AUTHORS	Wyatt, Nicola; Watson, Hannah; Anderson, Carl; Kennedy, Nicholas A; Raine, Tim; Ahmad, T.; Allerton, Dean; Bardgett, Michelle; Clark, Emma; Clewes, Dawn; Cotobal Martin, Cristina; Doona, Mary; Doyle, Jennifer; Frith, Katheirne; Hancock, Helen; Hart, Ailsa L.; Hildreth, Victoria; Irving, Peter; Iqbal, Sameena; Kennedy, Ciara; King, Andrew; Lawrence, Sarah; Lees, Charlie W.; Lees, Robert; Letchford, Laura; Liddle, Trevor; Lindsay, James; Maier, Rebecca; Mansfield, John; Marchesi, Julian R.; McGregor, Naomi; McIntyre, Rebecca; Ostermayer, Jasmin; Osunnuyi, Tolulope; Powell, Nick; Prescott, Natalie; Satsangi, J; Sharma, Shriya; Shrestha, Tara; Speight, Ally; Strickland, Michelle; Wason, James; Whelan, Kevin; Wood, Ruth; Young, Gregory; Zhang, Xinyue; Parkes, M; Stewart, Christopher; Jostins-Dean, Luke; Lamb, Christopher

VERSION 1 – REVIEW

REVIEWER	Kurada, Satya Indiana University School of Medicine
REVIEW RETURNED	08-May-2023

GENERAL COMMENTS	The concept of CD-MetaResponse is not very clear. I am assuming this is a nested case-control from cohort study. How will matching be done? This wasn't explained very well. Can the criteria for CD-meta response sub cohort be pre-defined?
---

REVIEWER	Nascimento, Roberto de Paula do University of Campinas Institute of Biology
REVIEW RETURNED	19-Jun-2023

GENERAL COMMENTS	The protocol is consistently well-written. A few suggestions are below. Introduction: 1. Please include epidemiological information not only from the UK. It is important to highlight where IBDs are becoming more prevalent/concerning, for example, in South America (first paragraph).2. Add brief information on the potential/promising utilization of bioactive compounds from plant products, based on clinical trials (second paragraph).
---

	Figure 1: please enlarge this figure (landscape may help). Necessary:  1. Highlight how this tool may help other people (not only in the UK), especially those in less developed areas/countries. 2. This protocol may take advantage of having a figure summarizing the main "problem" of the article with updated information on IBDs.
--	--

REVIEWER	Li, Jingsong Zhejiang University
REVIEW RETURNED	28-Jul-2023

GENERAL COMMENTS	The protocol is trying to build a tool aimed to predict treatment response of patients with Crohn's disease and ulcerative colitis based on multi-omics data. The detailed inclusion/exclusion criteria and definition of clinical outcome measures should be further reviewed by clinical experts majored in Crohn's disease and ulcerative colitis. There're still some issues could be improved.  1. The authors should cite more recent state-of-art research. 2. For sample size calculation, it would be better to firstly generate a minimum sample size by calculation rather than compared with similar studies. 3. The protocol should state more detailed about how to record and analysis the loss of follow-up. 4. As the protocol is trying to build a predictive tool aimed to predict treatment response using real-world data, it's recommended to reference the TRIPOD statement and FDA guidelines about real-world evidence. According to these document, the authors should provide more details about statistical methods for tool construction, especially for missing data processing. In addition, as the predictive model is built with microbiome, host genetic and clinical features, the authors may considering using more machine learning methods to find the most suitable modeling method for current study.
---

VERSION 1 – AUTHOR RESPONSE

Response to Reviewer 1, Dr Satya Kurada

2. The concept of CD-MetaResponse is not very clear. I am assuming this is a nested case-control from cohort study. How will matching be done? This wasn't explained very well. Can the criteria for CD-meta response sub cohort be pre-defined?

Response: Thank you to the reviewer for raising this query. CD-metaRESPONSE is a nested sub-cohort of 300 participants with Crohn's disease. Generous funding was provided by the Leona M. and Harry B. Helmsley Charitable Trust (funder reference 2002-04255). This funding will enable collection of additional detailed dietary intake data, and matched blood and stool samples from 300 participants at baseline and week 14. Additional dietary information collected will consist of a 4-day food diary questionnaire capturing all food and drink consumed during this period. Blood and stool samples will be used to undertake metabolomic profiling. The IBD-RESPONSE Study will recruit participants from (provisionally) 40 sites within the United Kingdom. A subset of 17 centres will initially preferentially recruit participants with Crohn's disease to the CD-metaRESPONSE sub-cohort. Once the recruitment target of 300 participants has been achieved, these sites will recruit participants with Crohn's disease to the main cohort only. The remaining sites will recruit participants with Crohn's

disease to the main cohort only. There are no additional inclusion criteria applicable to participants recruited to CD-metaRESPONSE.

Additional detail has been provided in paragraph 2 under the 'Study design' subheading (within the 'METHODS AND ANALYSIS' section) of the paper (page 9).

Response to Reviewer 2, Dr Roberto de Paula do Nascimento

3. Introduction: Please include epidemiological information not only from the UK. It is important to highlight where IBDs are becoming more prevalent/concerning, for example, in South America (first paragraph).

Response: Thank you for drawing this to our attention. We acknowledge that only limited epidemiological information was included in the original submission. We have commented on epidemiological trends elsewhere in the first paragraph of the introduction; "Outside of Western Europe and North America, the incidence is rising rapidly in many regions including South America, Latin America, Asia and Africa".

4. Introduction: Add brief information on the potential/promising utilization of bioactive compounds from plant products, based on clinical trials (second paragraph).

Response: We feel that specific discussion about potential use of bioactive compounds from plant products is beyond the scope of the introduction. We refer to potential alternative translational outputs of the study that may include non-pharmacological approaches to manipulate the host microbiome within the amended final paragraph on the study rationale (see response point 6).

5. Figure 1: please enlarge this figure (landscape may help).

Response: Figure 1 font sizes and the size of some of the illustrations within figure have been increased. The figure is provided as a 600DPI JPEG in landscape format.

6. Necessary: Highlight how this tool may help other people (not only in the UK), especially those in less developed areas/countries.

Response: We have included additional brief discussion of the potential non-pharmacological, translational outputs of this study. This can be found as the last paragraph under the 'Study rationale' heading, within the 'METHODS AND ANALYSIS' section of the paper;

"Through a multi-omic, precision medicine approach, the IBD-RESPONSE study seeks to improve selection of the right drug, for the right patient, at the right time. Other translational outputs of IBD-RESPONSE could bring into focus potential non-pharmacological approaches to treating IBD that do not necessarily involve large health economic expenditure. This could include manipulating the gut microbiome via the microbiota, through refinement of faecal microbial transplant protocols, use of pre- and probiotics, and dietary interventions."

The figure legend for the new Figure 1 (see response point 7) also refers to the potential to develop non-pharmacological interventions such as dietary modification through precision-medicine based approaches.

7. Necessary: This protocol may take advantage of having a figure summarizing the main "problem" of the article with updated information on IBDs.

Response: Thank you for this suggested addition. We have included an additional figure (Figure 1) to summarise the concept of precision medicine and how this may be compared to current (and

previous) approaches to treating complex diseases such as IBD. Figure 1 and 2 from the original submission have been renamed Figure 2 and 3, respectively.

Response to Reviewer 3, Dr Jingsong Li

8. The authors should cite more recent state-of-art research.

Response: To our knowledge, there is no more recently published data from large scale, well-powered studies evaluating treatment responses to multiple advanced therapies with or without independent replication cohorts. Detailed discussion of different applications of multi-omics in other clinical scenarios is not discussed as this is beyond the scope of the paper and the primary scientific objective of the IBD-RESPONSE study.

9. For sample size calculation, it would be better to firstly generate a minimum sample size by calculation rather than compared with similar studies.

Response: We thank the reviewer for the suggestion, and now include a supplementary figure and text (in the 'METHODS AND ANALYSIS' section of the paper, under the subheading 'Sample size calculations', page 17), describing the minimum sample size required to achieve 80% power under different assumptions.

10. The protocol should state more detailed about how to record and analysis the loss of follow-up.

Response: We thank the reviewer for drawing this to our attention. Loss to follow-up was discussed in various places. We have now brought it all together, along with discussion of missing data, and added more detail, in a new paragraph in the 'METHODS AND ANALYSIS' section of the paper under the 'Statistical analysis' sub-heading (page 22).

11. As the protocol is trying to build a predictive tool aimed to predict treatment response using real-world data, it's recommended to reference the TRIPOD statement and FDA guidelines about real-world evidence. According to these document, the authors should provide more details about statistical methods for tool construction, especially for missing data processing. In addition, as the predictive model is built with microbiome, host genetic and clinical features, the authors may considering using more machine learning methods to find the most suitable modeling method for current study.

Response: We now cite the TRIPOD checklist and have included information on the handling of missing data in the 'METHODS AND ANALYSIS' section of the paper, under the 'Statistical analysis' sub-heading (page 22). We have also included a section discussing sensitivity analyses using alternative predictive modelling/machine learning approaches.

VERSION 2 – REVIEW

REVIEWER	Li, Jingsong Zhejiang University
REVIEW RETURNED	26-Sep-2023
GENERAL COMMENTS	1. It seems current protocol did not restrict the medication of participants, which may result in hundreds of drug combinations. If so, the proposed protocol may not be able to build a sufficient model. Therefore, the authors should state the solution for such potential situation.

	2. In addition to AUROC, positive and negative prediction power, sensitivity and specificity should also be measured for the clinical predictive model. 3. For line 5 of page 34 of 112, dose the word "cencured" means "censored"?
--	---

VERSION 2 – AUTHOR RESPONSE

Response to Reviewer 3, Dr. Jingsong Li, Zhejiang University, Zhejiang Lab

1. It seems current protocol did not restrict the medication of participants, which may result in hundreds of drug combinations. If so, the proposed protocol may not be able to build a sufficient model. Therefore, the authors should state the solution for such potential situation.

Response: We thank the reviewer for raising this, and the general issue of the large number of potential clinical confounders in our study is one we have thought carefully about. We are carrying out an interim analysis of preliminary microbiome data to define which subset of clinical covariates, including drugs and drug combinations, independently explain significant amounts of microbiome variation in IBD patients and thus need to be included (either individually or in combination) as covariates in our main analyses. All other clinical covariates will be analysed in post-hoc sensitivity analyses. This will all be defined in our Statistical Analysis Plan, which will be produced prior to starting our primary, secondary and exploratory analyses.

2. In addition to AUROC, positive and negative prediction power, sensitivity and specificity should also be measured for the clinical predictive model.

Response: Sensitivity, specificity, positive and negative predictive power will be reported. We have clarified this on page 22, line 14-16 of the revised manuscript. Please also see response to reviewer comment 1.

3. For line 5 of page 34 of 112, dose the word "cencured" means "censored"?

Response: We thank the reviewer for highlighting this error - "censored" has been replaced with "censored".

VERSION 3 – REVIEW

REVIEWER	Li, Jingsong Zhejiang University
REVIEW RETURNED	30-Nov-2023
GENERAL COMMENTS	The propsed solution for the mentioned issue is acceptable as a protocol, but we still suggest the authors state a bit more detail in the statistic analysis section.